

# Geographic distributions and patterns of co-occurrence among black-bellied and shovel-nosed salamanders (*Desmognathus spp*.) in the Great Smoky Mountains National Park

Aidan Shaw, Rebecca Chastain and Benjamin M. Fitzpatrick

Department of Ecology and Evolutionary Biology, University of Tennessee—Knoxville, Knoxville, Tennessee, United States

## ABSTRACT

The southern Appalachian Mountains are a global hotspot for salamander diversity. Recent taxonomic revisions driven by a growing understanding of cryptic diversity and advancements in genomic technology have increased the number of described species in the area significantly, raising questions about biogeography and community structure. Recently described species in the *Desmognathus quadramaculatus-marmoratus* complex are morphologically cryptic but diagnosable by mitochondrial DNA. The complex includes multiple species within each of two distinct ecomorphs, the highly aquatic 'shovel-nosed' (SN) ecomorph and the semi-aquatic 'black-bellied' (BB) ecomorph. Here, we use mitochondrial DNA and Bayesian phylogenetic analysis to clarify distributions and patterns of co-occurrence of recently described species in the Great Smoky Mountains. We present new data showing more extensive co-occurrence of two cryptic species of the black-bellied ecomorph than previously recognized. Our results are also consistent with earlier work indicating shared variation between ecomorphs within one clade. In addition, we identified a divergent mitochondrial lineage of the shovel-nosed ecomorph related to *D. aureatus*, a species not previously known to inhabit the Great Smoky Mountains. These results reveal a more complex and diverse assemblage of ecomorphs than previously recognized in this hotspot of salamander diversity.

## INTRODUCTION

National parks and other protected areas are tremendous public resources and critical reservoirs of biological diversity in a time of unprecedented habitat destruction (*Hobbs et al., 2010*). Millions of visitors benefit from recreation and educational opportunities in U. S. National Parks every year (*Martin et al., 2011*). National Parks also serve as invaluable laboratories for natural science research (*Rodhouse, Sergeant & Schweiger, 2016*) and repositories of archeological and recent history (*Vukomanovic & Randall, 2021*). The

Corresponding author
Aidan Shaw,
aidanshaw0421@gmail.com

native species inhabiting parks and protected areas are an essential dimension of their value.

To understand the value of parks for biodiversity, it is critical to know which species occur in protected lands. This is an ongoing research need, in part because the scientific understanding of species boundaries and taxonomy is continually evolving. For example, recent changes in salamander taxonomy have resulted in a marked increase in the number of recognized species in southeastern North America (*Tilley et al., 2013*; *Camp & Wooten, 2016*). The Great Smoky Mountains National Park (GSMNP) is a hotspot of salamander diversity (*Dodd, 2004*), but the number and identity of the species occurring there is uncertain after recent taxonomic revisions of the genus *Desmognathus* (*Pyron & Beamer, 2022*, *2023*). Here we attempt to clarify the distributions and patterns of co-occurrence of three recently described species with respect to the GSMNP.

'Black-bellied' and 'shovel-nosed' salamanders (BB and SN) are the two most aquatic ecomorphs in *Desmognathus* (*Petranka, 1998*; *Bruce, 2011*). Their ecological syndromes are evident in morphological comparison (Fig. 1). Black-bellied salamanders are the less aquatic of the two. They have round, bulging eyes and a heavily keeled tail that terminates in a point. They are the largest ecomorph in the genus, with adults reaching >100 mm SVL (*Valentine, 1974*). Shovel-nosed salamanders are the more aquatic ecomorph and are characterized by lower-profile, almond-shaped eyes and a heavily keeled, spatulate tail (*Martof, 1962*). Shovel-nosed salamanders are also large compared with other *Desmognathus spp.*, with adults reaching >70 mm SVL (*Martof, 1962*). Both ecomorphs have darkly pigmented venters and dorsal patterns vary from nearly black to a greenish speckling common to SN or a light orange-brown common to BB (*Martof, 1962*; *Niemiller & Reynolds, 2011*). Larvae are difficult to identify at early life stages but become more easily distinguishable near time of metamorphosis (*Niemiller & Reynolds, 2011*). Large BB larvae are often olive green in color with paired orange-yellow dorsal spots, while large SN larvae are usually a significantly darker, black-velvet color with distinctly white gills and spatulate tails (*Martof, 1962*).

Historically, BB and SN were recognized as two species (*D. quadramaculatus* Holbrook 1840 and *D. marmoratus* Moore 1899); however, recent molecular systematics has mapped the two ecomorphs to two deeply divergent clades within *Desmognathus* (*Jackson, 2005*; *Jones & Weisrock, 2018*) and subsequently split each ecomorph into multiple species (*Pyron et al., 2022*; *Pyron & Beamer, 2022*, *2023*). *Jackson (2005)* recognized a northern clade containing BB and SN from the northern extent of their range in Virginia south through the Blue Ridge to roughly midway through Tennessee and North Carolina, and a southern clade ranging from contact with the northern clade through North Georgia and South Carolina (see Fig. 2 for previous sampling of each clade). Subsequent studies (*Jones & Weisrock, 2018*; *Pyron et al., 2022*) have denoted these clades as the Pisgah (northern) and Nantahala (southern) clades. Both *Jackson (2005)* and *Jones & Weisrock (2018)* detected introgression between BB and SN ecomorphs in the Pisgah clade but no evidence of introgression between ecomorphs in the Nantahala clade, and it seems clear that there is no gene flow between Pisgah and Nantahala lineages (*Pyron et al., 2020*).

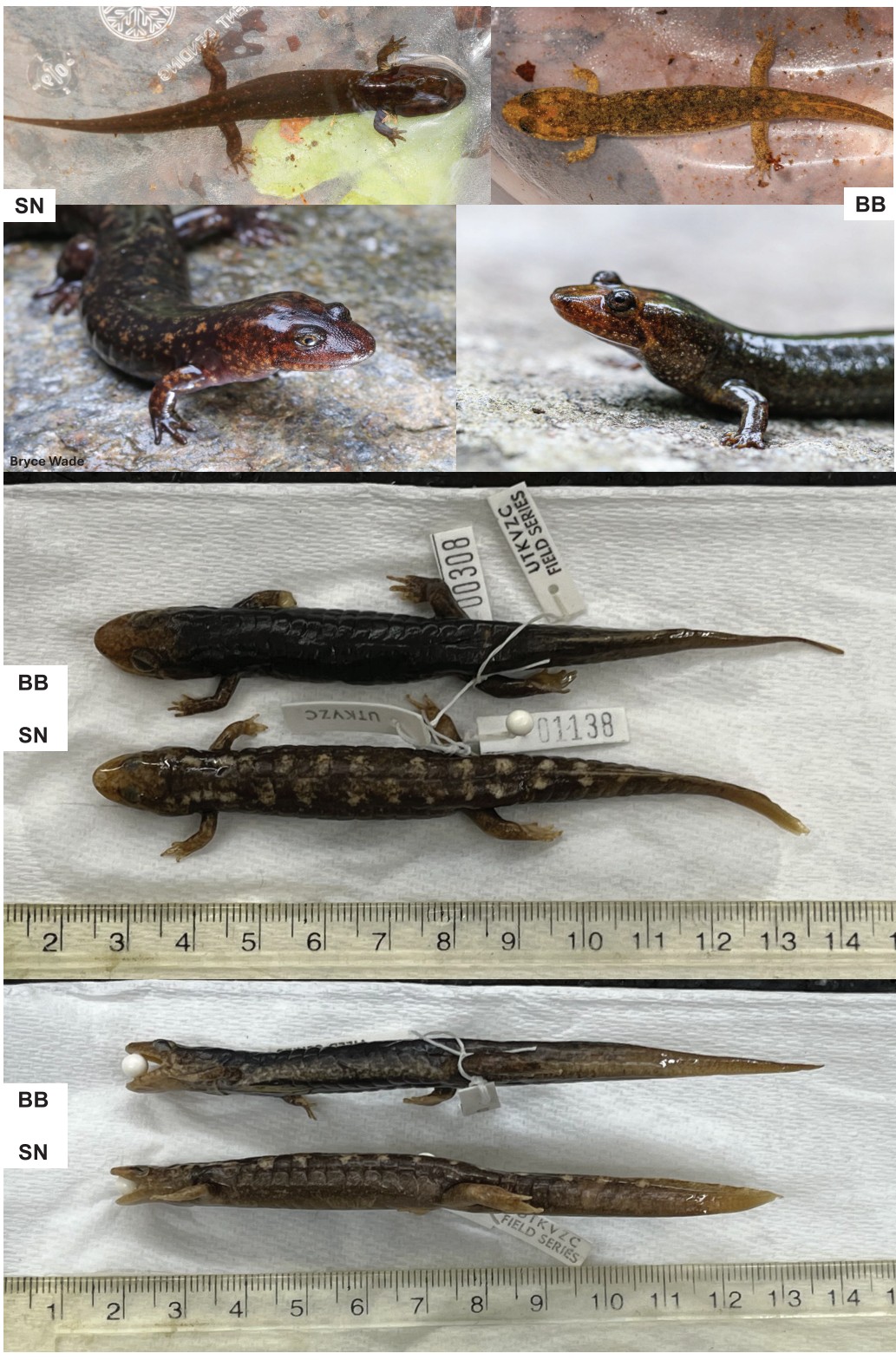

**Figure 1 Morphological comparison of shovel-nosed and black-bellied ecomorphs.** Note variation in larval color, head shape and posture, and tail shape (top to bottom). Photo credits AS, BMF, and Bryce Wade. Specimens are from the UTK zoological teaching collection.

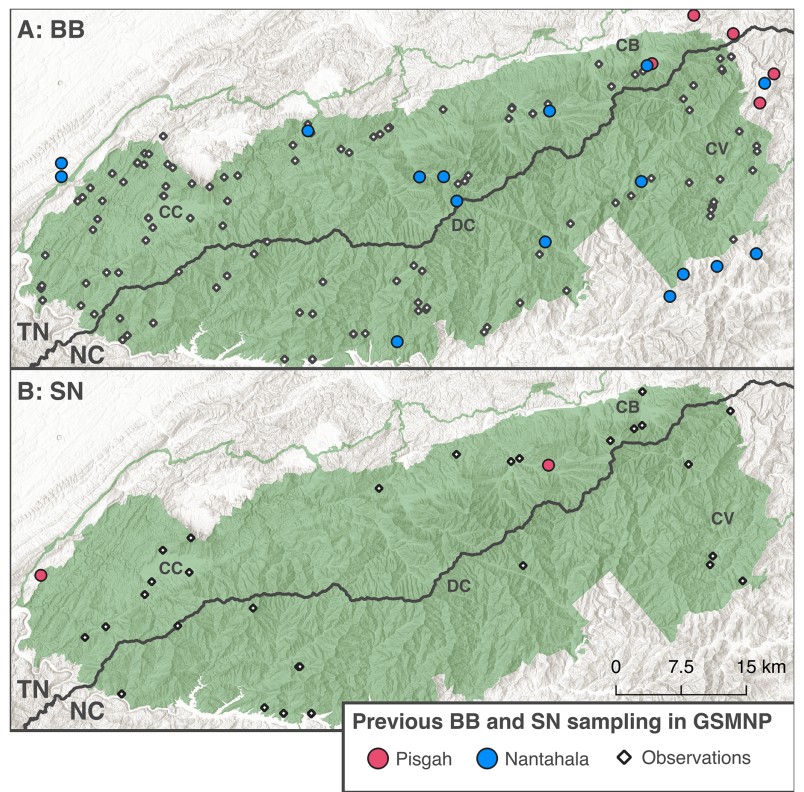

**Figure 2 Distribution of existing BB and SN sampling in the GSMNP.** The round points indicate Black-bellied (A) and Shovel-nosed (B) salamander samples from *Jackson (2005)*, *Beamer & Lamb (2020)*, *Jones & Weisrock (2018)*, *Pyron et al. (2020)*, and *Pyron et al. (2025)*. Colors indicate clades reported by original study. The black diamonds denote observational data for each ecomorph from *Dodd (2004)*. Basemaps copyright *National Park Service (2024)* and *Esri (2024)*, map produced using *QGIS Geographic Information System (2024)* (v3.40.3).

Co-occurrence of divergent clades of the same ecomorph has rarely been documented in previous studies, with only one site in each of *Jackson (2005)*, *Jones & Weisrock (2018)*, *Pyron & Beamer (2022)*, and *Pyron et al. (2025)* where both clades of black-bellied salamanders occurred together or in the same local stream system. There have been no documented sites where shovel-nosed salamanders from both clades co-occur. This pattern supports the hypothesis that co-occurrence and divergence of BB and SN ecomorphs has been facilitated by ecological niche partitioning. Based on our understanding of the ecological roles of BB and SN salamanders, competitive exclusion ought to preclude extensive co-occurrence of different species of the same ecomorph (*Hardin, 1960*; *Hairston, Nishikawa & Stenhouse, 1987*; *Bruce, 2011*). Divergence between species of each ecomorph is strongly influenced by watershed connectedness, especially in the more obligately aquatic SN, with apparently ancient isolation between populations on either side of the Eastern Continental Divide (*Voss et al., 1995*; *Jackson, 2005*).

The GSMNP includes the type locality of one newly described Nantahala clade BB: *D. gvnigeuswotli* (*Pyron & Beamer, 2022*). Based on limited genetic sampling, *D. gvnigeuswotli* is thought to occur throughout the mountain range along with one Pisgah

clade SN: *D. intermedius* (*Pyron & Beamer, 2023*). One record indicates the presence of one Pisgah clade BB (*D. mavrokoilius*) in the northeastern part of the park (*Jackson, 2005*). Here we use expanded sampling in the GSMNP to document co-occurrence between black-bellied salamanders from two deeply divergent clades (*D. mavrokoilius* and *D. gvnigeuswotli*), corroborate previous reports of a lack of genetic differentiation between co-occurring black-bellied and shovel-nosed ecomorphs within the Pisgah clade (*D. mavrokoilius* and *D. intermedius*), and present preliminary data indicating the presence of an additional clade of shovel-nosed salamanders in Cades Cove, GSMNP.

## METHODS

### Sampling

We sampled BB and SN salamanders throughout the GSMNP starting in summer 2023. As part of an ongoing research program to genetically evaluate previously documented populations, we visited sites with known or suspected occurrences of BB or SN, with an emphasis on sites with co-occurrence of both ecomorphs (*Dodd, 2004*). In addition, many BB and some SN were sampled opportunistically in smaller seeps and creeks throughout the GSMNP. At six sites, we obtained large population samples as bycatch during three-pass electrofishing brook trout surveys carried out by park service scientists. We released each individual alive after removing roughly 4–10 mm of tail tissue for DNA analysis. We identified each adult as BB or SN based on eye, head, and tail shape and we identified larvae by color and tail shape (*Martof, 1962*; Fig. 1). We also photographed most individuals for reference. Tissues were suspended in either ethanol or salt lysis buffer and stored at −20C until used. Due to our non-lethal sampling technique, we usually sampled every individual caught, except for very small larvae (less than 2.5 cm total length) and individuals with prior tail damage. This provided large sample sizes from multiple major creeks and watersheds throughout the GSMNP (Fig. 3). Tissue sampling was carried out in compliance with the United States Animal Welfare Act [7 U.S.C. 2131 et seq.] and according to University of Tennessee Institutional Animal Care and Use Committee (IACUC) protocol 2,710 and National Park Scientific Research Collection Permit GRSM-2023-SCI-2209. See the methods appendix in the Supplemental Material for more information on sampling.

### Molecular preparation and phylogenetic analysis

For species level identification, we sequenced a portion of the mitochondrial cytochrome B gene and compared our results to the extensive reference dataset generated by *Jackson (2005)*. We extracted DNA from tail tissues using either a DNeasy Blood and Tissue kit (Qiagen, Hilden, Germany) or salt extraction (*Sambrook & Russell, 2001*). We performed PCR to amplify approximately 500 bp of cytochrome B using primers and protocol adapted from those described by *Jackson (2005)*. PCR products were purified using ExoSAP-IT and sequenced by Eurofins Genomics (Louisville, KY, USA). See the methods appendix for more information on DNA methods.

We trimmed our sequences to account for declining quality at the ends and assembled consensus sequences from forward and reverse reads. We then combined our sequence

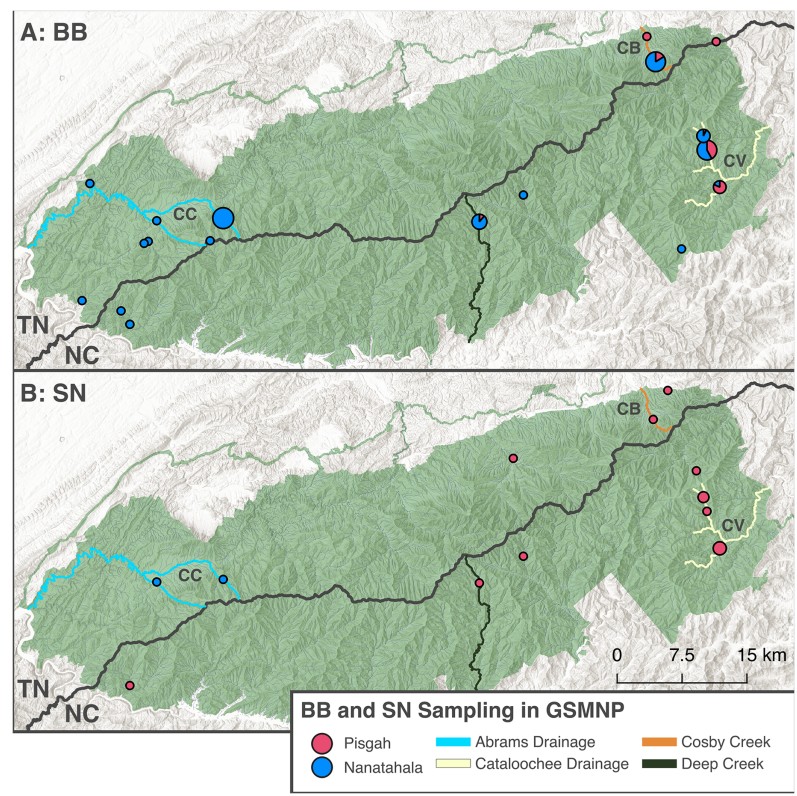

**Figure 3** **Distribution of samples from the GSMNP that contributed to this study.** The pie chart size is scaled according to sample size for a given locality and colored based on mtDNA clade assignment. Samples are split between black-bellied (A) and shovel-nosed (B) ecomorphs. 'Mixed features' individuals are not included in this figure. Regions of the GSMNP which are of interest to this study are labelled (CC) Cades Cove, (CB) Cosby, and (CV) Cataloochee Valley. The major stream systems that are referred to in this article are highlighted, including the Abrams Creek Drainage (which includes Anthony and Mill Creeks), the Cataloochee River Drainage (including Onion Bed Branch, Rough Fork, and Pretty Hollow Creek), Cosby Creek, and Deep Creek. Basemaps copyright *National Park Service (2024)* and *Esri (2024)*, map produced using *QGIS Geographic Information System (2024)* (v3.40.3).

data with 373 sequences generated by *Jackson (2005)* and performed phylogenetic analyses using two Bayesian inference models. A codon-partitioned analysis in MrBayes (v3.2.7) was carried out according to the specifications described by *Jackson (2005)*, and a similar analysis was performed in BEAST (v1.10.4) using the Yang96 site partitioning model (*Yang, Cheng & Kain, 1996*; *Ronquist et al., 2012*; *Suchard et al., 2018*). We compared the topology of the maximum clade credibility trees from each model and topology reported by previous literature.

## Comparison of ecomorphs within the northern clade

To test the hypothesis of genetic differentiation between BB and SN ecomorphs in the Pisgah clade, we estimated genetic differentiation between each morph and sampling site ($\varphi_{ST}$) using pairPhiST from the package haplotypes (v1.1.3.1) using a single factor to label population and ecomorph (*Aktas, 2015*; *R Core Team, 2024*). We also performed a

distance-based redundancy analysis (dbRDA) using the function capscale in vegan (v2.6-8) on the raw sequence divergence matrix and performed model selection to determine whether population and ecomorph are significant predictors of genetic divergence (*Oksanen et al., 2024*). For these analyses, we focused only on sites for which we had more than 10 Pisgah clade individuals.

## RESULTS

Here we report results from mtDNA data obtained from a sample of 270 individuals from 27 sites in the GSMNP (Fig. 3). Sample sizes for each site ranged from $n = 1$ to 48 individuals, $\bar{x} = 10.0$. We collected DNA from 163 BB, 83 SN, and 24 individuals that were not identified in the field. We identified salamanders in the field using eye, head, and tail shape as well as larval color as described above. Morphologically ambiguous individuals were described qualitatively or marked as 'mixed features.' Individuals marked as 'mixed features' most often had BB-like faces and SN-like spatulate tails, all of which had Pisgah clade mtDNA.

### Phylogenetic analysis

Processing of sequence data resulted in an alignment of 375 bp which was combined with the (*Jackson, 2005*) mtDNA population set for phylogenetic analysis. The topology of major clades within *Desmognathus* was consistent between MrBayes and BEAST analyses and with those reported by previous research. Clade assignment of our samples was consistent across both models, and largely consistent with field identifications and expected geographic ranges (Fig. 4).

Six individuals from two sites in Cades Cove, GSMNP (denoted "CC" in Fig. 3) that were identified in the field as SN grouped as sister to the Nantahala clade containing *D. folkertsi* and *D. aureatus* with strong support in both analyses (0.99–1.0). Of these six SN, three were sampled from Mill Creek at the western end of Cades Cove and three from Anthony Creek at the eastern end (see Fig. 3). Neither the dwarf black-bellied salamander, *D. folkertsi*, nor the Southern SN (*D. aureatus Pyron & Beamer, 2023*) have been reported in the Great Smoky Mountains or anywhere in Tennessee. Our Cades Cove SN clade is strongly supported as the sister group to the clade containing *D. aureatus* and *D. folkertsi* (Fig. 4), but are not included within *D. aureatus*, also with strong support (0.90–0.99). One sample from Cades Cove (from Mill Creek on the west end of the Cove) was uncertain in its relationship to other Cades Cove samples. We represent this with a polytomy that we are unable to resolve with our limited sequence data. The branch lengths estimated by BEAST indicate divergence between the Cades Cove clade and the clade containing *D. aureatus* and *D. folkertsi* being similarly ancient to that between the two described Nantahala BB species (*D. gvnigeusgwotli* and *D. amphileucus*).

All other SN sequences clustered unambiguously with the Pisgah clade, and all BB sequences clustered unambiguously with either the Pisgah clade or *D. gvnigeuswotli* within the Nantahala clade, except for seven samples that were excluded from *D. gvnigeuswotli* with weak support (0.3) but still grouped with Nantahala BBs (Fig. 4). These seven grouped with strong support (0.99) with a BB from *Jackson's (2005)* dataset that was collected along

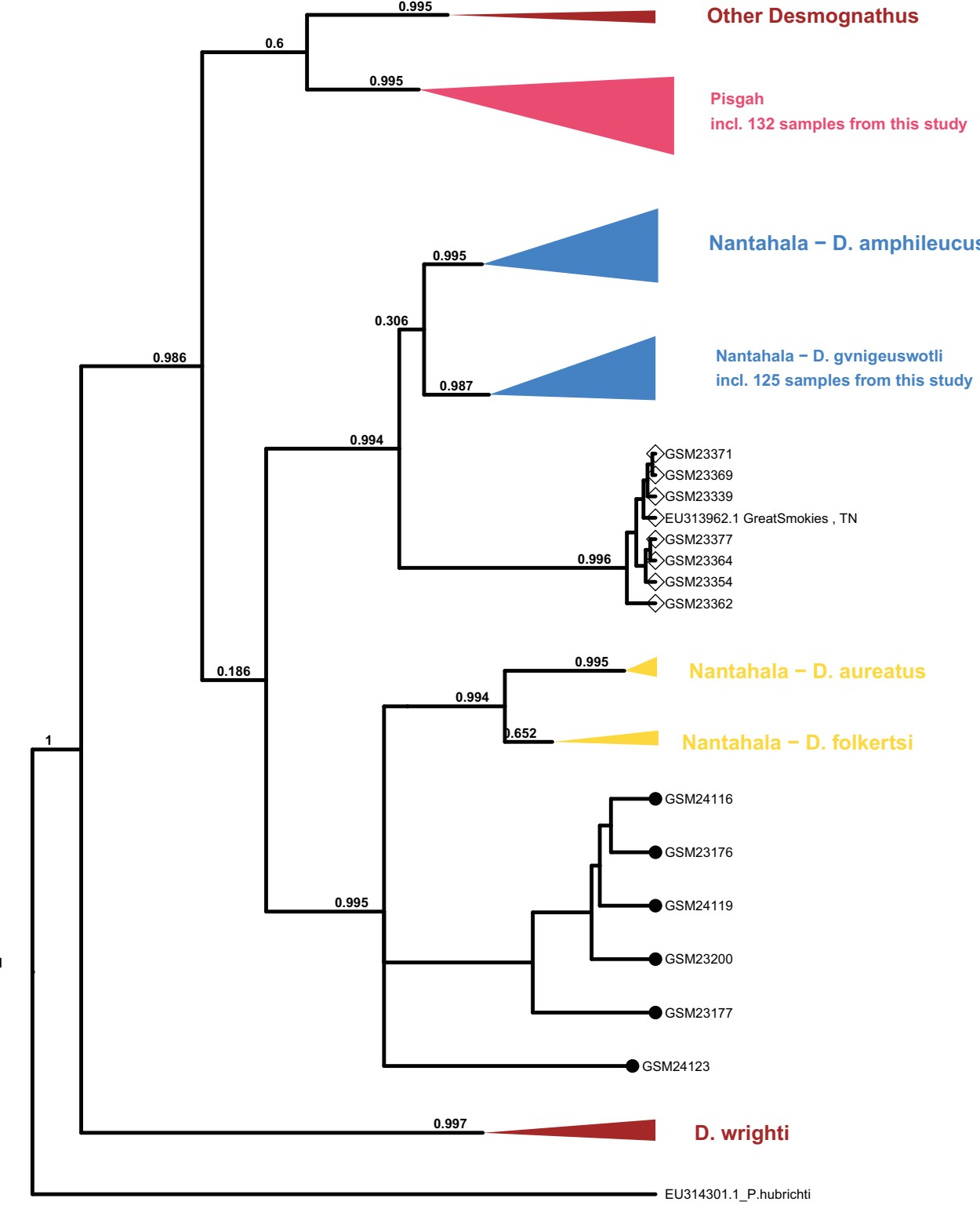

**Figure 4 Collapsed maximum clade credibility (MCC) tree from BEAST.** The clade posterior support is in black. Samples new to this study are designated with numbers beginning with "GSM." The tip points denote ecomorph with SN represented by closed circles and BB by open diamonds. Data from GenBank are designated with their accession numbers and brief locality information. *Phaeognathus hubrichti* is the outgroup. The median log-likelihood for the BEAST tree was −7,293.5.

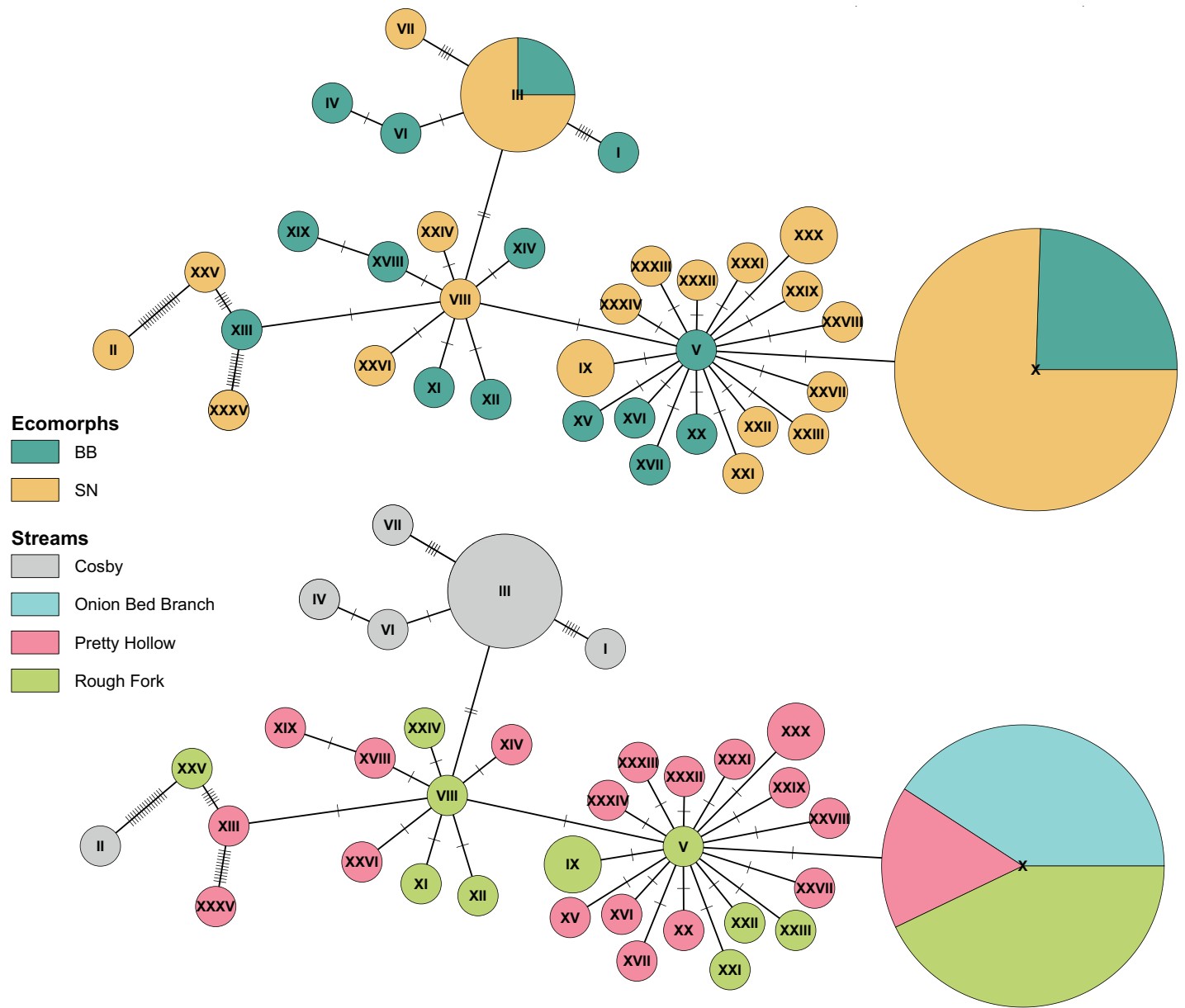

**Figure 5  Haplotype network of Pisgah clade BB and SN from four sites with population samples.** The statistical parsimony haplotype network was created with the function haploNet from *pegas* (v1.3, *Paradis, 2010*). The top and bottom networks are identical, with different colors to compare grouping of ecomorphs with grouping of localities. The size of the pie is scaled by the number of samples with each haplotype. The most frequently observed haplotype (**X**) represents 49 individuals, (**III**) represents 8, and most of the rest are singletons. Each score on the connecting lines represents one SNP between haplotypes. Note the shared haplotypes among BB and SN in the top representation and the rough grouping by locality in the bottom.                                                        

US 441 in the GSMNP, and the topology of the Nantahala BB clade reflects that of the cyt B tree reported by Jackson. Within the Pisgah clade, our sequences clustered with specimens that would be classified as *D. mavrokoilius* (if BB) or *D. intermedius* (if SN) according to the taxonomy of *Pyron & Beamer (2022, 2023)*. As with previous results, there was no

**Table 1 Count of each species identified at each sampling site.** Species identifications are based on field identification of ecomorph and mtDNA clade assignment. Individuals that were not confidently identified in the field are listed in the "Ecomorph Uncertain" column.

| Sample sites | D. cf. aureatus | D. gvnigeusgwotli | D. intermedius | D. mavrokoilius | D. monticola | Pisgah clade mixed features | Ecomorph uncertain | Site totals |
|---|---|---|---|---|---|---|---|---|
| Anthony Creek 3-pass | 3 | 35 | | | | | | **38** |
| Bird Branch on Old Settlers Trail | | | 1 | | | | | **1** |
| Bunches Creek | | 1 | | | | | | **1** |
| Campsite 95, Wolf Ridge Trail | | 3 | | | | | | **3** |
| Cosby Creek 3-pass 1 | | 9 | 6 | 3 | | | 3 | **21** |
| Cosby Creek nature trail | | 1 | | | | | | **1** |
| Cosby Creek-Low Gap trail | | 1 | | | | | | **1** |
| Cosby Creek 3-pass 2 | | 14 | 2 | 2 | | | | **18** |
| Davenport Gap Shelter | | | | 1 | | | | **1** |
| Deep creek 3-pass 1 | | 8 | 4 | 1 | | | 2 | **15** |
| Deep creek 3-pass 2 | | 8 | | 1 | | | | **9** |
| Kephart prong near shelter | | 1 | 1 | | | | | **2** |
| Kingfisher Creek | | 2 | | | | | | **2** |
| Mill Creek 3-pass | 3 | 3 | | | 1 | | | **7** |
| Gregory Ridge Trailhead | | 2 | | | | | | **2** |
| Onion Bed Branch 3-pass | | 13 | 19 | 1 | | | | **33** |
| Parson Branch Road first ford | | 1 | | | | | | **1** |
| Parsons Branch Road, Forge Creek | | 1 | | | | | | **1** |
| Pretty Hollow 3-pass | | 18 | 14 | 13 | | 3 | | **48** |
| Rough Fork 3-pass | | 2 | 23 | 9 | | 10 | | **44** |
| Rowdy Creek just above Hwy 32 | | | 1 | | | | | **1** |
| Swallow Fork | | | 1 | | | | | **1** |
| Tritt cemetery | | | | 1 | | | | **1** |
| Twentymile parking area | | 1 | | | | | | **1** |
| Twentymile 3-pass | | 3 | 5 | | | | 5 | **13** |
| Russell Field Shelter | | 1 | | | | | | **1** |
| Wolf Ridge Trail | | 3 | | | | | | **3** |
| **Species totals** | **6** | **131** | **77** | **32** | **1** | **13** | **10** | **270** |

**Table 2 Pairwise $\varphi_{ST}$ estimated from cytochrome b sequences using *haplotypes*.**

|  | Cosby BB | Cosby SN | Rough fork BB | Rough fork SN | Pretty hollow BB | Pretty hollow SN |
|---|---|---|---|---|---|---|
| Cosby SN | 0.000 |  |  |  |  |  |
| Rough Fork BB | 0.560* | 0.357* |  |  |  |  |
| Rough Fork SN | 0.584* | 0.455* | 0.011 |  |  |  |
| Pretty Hollow BB | 0.468* | 0.343* | 0.135* | 0.130* |  |  |
| Pretty Hollow SN | 0.423* | 0.332* | 0.107* | 0.109* | 0.000 |  |
| Onion Bed Branch SN | 0.735* | 0.521* | 0.090* | 0.028 | 0.265* | 0.203* |

**Note:**
* indicates $p < 0.05$ based on 10,000 permutations.

consistent mtDNA differentiation between BB and SN ecomorphs within the Pisgah clade (Fig. 5).

## Patterns of co-occurrence

Congruent with previous results, our data (Table 1) support existing knowledge that co-occurrence of BB and SN ecomorphs in and around streams is common (*Martof, 1962*; *Martof & Scott, 1957*; *Petranka, 1998*; *Dodd, 2004*). BB and SN occurred together in 10 of our 27 sites. Of the remaining 17 sites, 14 were BB-only sites. This highlights the pattern that BB are less dependent on large streams than SN and more likely to be found in smaller seeps and trailside (*Dunn, 1926*; *Petranka, 1998*; *Bruce, 2011*). Additionally, in contrast with trends from previous work, we detected frequent co-occurrence of BB ecomorphs from both the Pisgah and Nantahala clades. In six sites on the eastern side of GSMNP, we found Pisgah SN and both Pisgah and Nantahala BBs. These were two sites in Cosby Creek (upstream and downstream of campground), three sites in the Cataloochee Valley (Pretty Hollow, Rough Fork, and Onion Bed Branch), and possibly in Deep Creek where two individuals out of 18 BB's had Pisgah mtDNA (Table 1; Fig. 3). All other sites (except for Cades Cove, see phylogenetic analysis) were characterized by co-occurrence of Pisgah SN (*D. intermedius*) and Nantahala BB (*D. gvnigeusgwotli*) or only one ecomorph.

## Genetic similarity within the Pisgah clade

Finally, also in agreement with previous results (*Jackson, 2005*), we found no consistent differentiation between Pisgah BB and SN. Ecomorphs that grouped with the Pisgah clade did not map to monophyletic groups and there were haplotypes shared between BB and SN in both Cosby and the Cataloochee Valley (Fig. 5). Genetic grouping was statistically associated with geography. Specifically, our samples from Cosby Creek and the Cataloochee Valley drainage in the GSMNP grouped with a large subset of *Jackson's (2005)* samples from TN and NC, including those from GSMNP and the Balsam Mountains, NC, with moderate posterior support (0.75). We estimated pairwise genetic divergence using four populations with large samples of Pisgah BB and SN (Cosby Creek, Rough Fork, Pretty Hollow, and Onion Bed Branch). The average $\varphi_{ST}$ between BB and SN from the same population was 0.004 and none were statistically significant (Table 2). The average $\varphi_{ST}$ between populations was 0.185 for BB and 0.275 for SN, with the largest values

**Table 3 Results of model selection performed on the dbRDA.**

| Model | DF | AIC |
|---|---|---|
| Full model | 5 | 244.53 |
| Populations only | 3 | 243.88 |
| Ecomorphs only | 1 | 258.55 |

**Table 4 Marginal effects of predictors of cytochrome b sequence divergence in the dbRDA.**

| | F | P |
|---|---|---|
| Populations | 3.59 | <0.001 |
| Ecomorphs | 1.46 | 0.262 |

**Note:**
P-values were estimated from 100,000 permutations of each factor while holding the other constant.

between Cosby and the three sites within the Cataloochee Valley. Model selection with the dbRDA indicates that the best model to describe variation within the Pisgah clade is one with only population as a predictor (Table 3). MtDNA variation was not statistically associated with ecomorph when tested alone or jointly with sample site as a factor (Table 4).

## DISCUSSION

The Great Smoky Mountains harbor some of the most diverse assemblages of salamanders anywhere in the world (*Petranka, 1998*; *Dodd, 2004*). Understanding the historical and contemporary processes responsible for the evolution and maintenance of such diversity depends on clear documentation of species distributions and patterns of co-occurrence. *Desmognathus* salamanders are a noted example of ecological niche partitioning, with morphological differences among species associated with differential use of aquatic and terrestrial microhabitats (*Bruce, 2011*). But they are also notorious for cryptic diversity and fuzzy species boundaries (*Tilley et al., 2013*; *Camp & Wooten, 2016*). Here we describe extensive co-occurrence of genetically distinct but morphologically indistinguishable BB salamanders, interbreeding between morphologically distinct but genetically similar BB and SN salamanders, and an undescribed mitochondrial lineage of SN salamanders that might be restricted to a small watershed in the northwestern corner of the Great Smoky Mountains.

Our results largely corroborate data from *Jackson (2005)*, *Jones & Weisrock (2018)*, and *Pyron et al. (2022)* with respect to phylogenetic relationships between the many cryptic lineages of BB and SN. Our finer-scale sampling throughout the GSMNP suggests that co-occurrence of BB from divergent clades is more extensive than previous data indicated. Frequent co-occurrence of cryptic lineages in a system where co-occurrence is understood as being facilitated by ecological niche partitioning is a challenge to the prevailing dogma (*Hairston, 1987*). Three potential explanations for this pattern are (1) that BBs from divergent clades partition habitat more finely than previously understood, (2) that some external disturbance or biotic interaction prevents competitive exclusion of one clade by

the other, or (3) co-occurrence is a temporary phase in the invasion and displacement of one species range by the other. Developing more detailed descriptions for BB microhabitat occupancy and resource use could help to address the first possibility. *Pyron & Beamer (2022)* showed subtle statistical differences in body proportions that might be associated with resource use. In future sampling efforts, care should be taken to record specific information such as stream size, cover object types, and water flow. For example, *Pierson, Fitzpatrick & Camp (2021)* found a difference in *Eurycea wilderae* and *E. cirrigera* microhabitat use by categorizing sections of stream as 'pools,' 'riffles,' or 'runs' and genotyping the salamanders that were caught in each flow type. A similar approach involving BB and SN ecomorphs could be helpful in building a higher resolution understanding of ecological strategies used by each ecomorph and cryptic lineage.

The lack of mtDNA differentiation between Pisgah BB and SN (*D. mavrokoilius* and *D. intermedius*) also corroborates previous data, adding to the need for population genetic analyses to resolve questions of species delineation and hybridization. *Jackson (2005)* concluded that mitochondrial and nuclear variation were associated with geography but not morphology within the Pisgah clade. *Pyron et al. (2020)* suggested that the Pisgah clade could be subdivided into several BB and SN species with repeated parallel evolution explaining the lack of correspondence between genomic and phenotypic similarity. Subsequently, *Pyron et al. (2022)* supported this interpretation with a larger dataset, but also presented evidence of mixed ancestry across several geographically adjacent groups within the Pisgah clade, consistent with hybridization between ecomorphs. *Pyron et al. (2025)* interpreted the lack of correspondence between ancestry and phenotype as the result of relatively rare, ancient hybridization coupled with a hypothetical quantitative genetic threshold mechanism determining expression of discrete phenotypic syndromes.

The above studies were broad in geographic scope, with rarely more than two specimens per locality. Our analysis of population samples found no mtDNA differentiation between ecomorphs living in the same 100 m reaches of Cosby Creek, Pretty Hollow Creek, and Rough Fork. The variation in our data is shared between ecomorphs (Fig. 5), with different ecomorphs in the same stream consistently more similar to each other than to their corresponding ecomorphs in other streams as close as 4.5 km away (Rough Fork to Pretty Hollow; Table 2). We fail to reject the null hypothesis of no differentiation between Pisgah clade phenotypes within streams. However, any single locus might be misleading, and our sample sizes are too modest to detect correlations between ancestry and phenotype much weaker than about 0.64 (*Cohen, 1988*). Our future work will characterize potential admixture between BB and SN and measure dispersal and geneflow between local populations. These data will be essential for understanding the ecological mechanisms that maintain BB and SN phenotypes and/or facilitate hybridization between the two.

Our detection of a previously undescribed SN lineage in Cades Cove, GSMNP highlights the pervasiveness of cryptic diversity in *Desmognathus* and warrants further study to determine its relationship to known southern shovel-nosed salamanders (*D. aureatus*), describe any distinctive morphological characteristics, and determine the extent of its geographic range. Six individuals from two sites in Cades Cove grouped as a sister clade to *D. aureatus* and *D. folkertsi*, and BEAST estimated relatively deep divergence, similar to

that between distinct Nantahala BB species. Moreover, the morphological and genomic distinctiveness of *D. folkertsi* (*Camp et al., 2002*) argues against classifying the Cades Cove clade as *D. aureatus* if additional molecular data corroborate its phylogenetic position as sister to *D. aureatus* and *D. folkertsi* (Fig. 4).

Our mtDNA analysis was designed for efficient categorization of samples into previously described clades (*Jackson, 2005*), not for *de novo* phylogenetic analysis. With only 375 base pairs, we replicated the major features of the previous analyses, but with less certainty and a few relatively minor differences in tree topology. From the 270 samples included in the phylogenetic analysis, a subset of 91 identified as Pisgah clade BB or SN were included in the haplotype network and genetic distance analyses.

Of the 270 individuals included in our phylogenetic analysis, only the six Cades Cove clade samples and one Nantahala BB were not within 93% sequence similarity to previously published haplotypes. Additionally, within our six Cades Cove SN samples, sequences were highly variable. The five samples that formed the monophyletic clade displayed up to 7% pairwise divergence, and GSM23123 was 9.8% different from the next closest Cades Cove sample. This high degree of variation and their being only 84% similar to the next closest species (*D. folkertsi*) on GenBank casts doubt on them being valid cytochrome B sequences. It is possible that these irregular sequences represent nuclear-mitochondrial insertions (NUMTs) rather than valid cytochrome B sequences. However, BLASTing our sequences against an existing genome assembly for *D. fuscus* returns no significant similarities, arguing against this possibility (GenBank acc.: GCA_050004315.1) Therefore, our diagnosis of the Cades Cove clade remains highly provisional until more comprehensive molecular analyses are completed.

In contrast to our findings of co-occurrence of BB ecomorphs from divergent clades, we found no evidence here of co-occurrence of Pisgah SN (*D. intermedius*) and our 'Cades Cove' SN (*D. cf. aureatus*). No Pisgah clade haplotypes were present in any samples from either Mill Creek or Anthony Creek, our two sites within Cades Cove. Both of our sampled creeks drain north into Abrams Creek, which flows west out of the Great Smoky Mountains (Fig. 3). The nearest documented SN (*D. intermedius*) was collected by *Jackson (2005)* in a creek that joins Abrams Creek lower in the watershed several miles below Abrams Falls. We also documented *D. intermedius* in the Twentymile Creek watershed, which is south of Cades Cove and hydrologically connected to Abrams Creek *via* the Little Tennessee River. This apparent lack of co-occurrence of SN species is consistent with the hypothesis that competitive exclusion impedes the coexistence of identical ecomorphs of divergent clades and further emphasizes the need for a better understanding of what might facilitate coexistence of distinct BB species. The presence of a Nantahala clade SN in a tributary of the Little Tennessee River in the GSMNP is especially shocking considering the relatively ancient divergence between SN on either side of the Eastern Continental Divide (*Voss et al., 1995*; *Jones et al., 2006*). *Jones et al. (2006)* suggested, based on a molecular clock analysis and geologic history of the region, that a geologic event isolated populations of SN, which are less likely than BB to disperse over land, before the Pleistocene. We hope that further sampling in Cades Cove and surrounding areas will clarify the geographic extent and evolutionary history of this disjunct lineage.

## CONCLUSIONS

Our results reveal a more complex and diverse assemblage of BB and SN *Desmognathus* in the Great Smoky Mountains National Park than previously recognized, emphasizing the value of parks for preserving biodiversity, both known and not yet described. The Cherokee Black-bellied Salamander (*D. gvnigeuswotli Pyron & Beamer, 2022*) appears to occur throughout the mountain range with the possible exception of some of the easternmost streams flowing directly into the Pigeon River. The Blue Ridge Black-bellied Salamander (*D. mavrokoilius Pyron & Beamer, 2022*) is common in the Cosby and Cataloochee watersheds in addition to smaller drainages such as Tobes Creek on the eastern end of the Great Smoky Mountains. It might occur sporadically as far west as Deep Creek. These two BB species cannot be distinguished with certainty by known visual characteristics (*Pyron & Beamer, 2022*). The central shovel-nosed salamander (*D. intermedius Pyron & Beamer, 2023*) occurs across the Great Smoky Mountains from the easternmost streams to the Twentymile Creek area in the southwest. Hybridization between *D. intermedius* and *D. mavrokoilius* might be common where they co-occur. We observed several individuals with Pisgah clade mtDNA with the head and eyes of the BB morph but the spatulate tail of the SN morph. We also detected a potentially novel mitochondrial lineage of SN salamanders in Cades Cove. Further research is needed to determine the phylogenetic composition of SN salamanders in the north-western quadrant of the Great Smoky Mountains.

## ACKNOWLEDGEMENTS

We would like to thank research coordinator Paul Super as well as Matt Kulp, Caleb Abramson, and the fishery science team with the National Parks Service for facilitating our access to the GSMNP and letting us join electrofishing surveys. We thank Rebecca Smith for her assistance in sampling and consistent support. We also thank Jon Davenport along with the Appalachian State Herpetology class for their assistance in sampling.

### Funding

This work was supported by a Student Faculty Research Award to Rebecca Chastain and Benjamin Fitzpatrick by the University of Tennessee Graduate School. The funders had no role in study design, data collection and analysis, decision to publish, or preparation of the manuscript.

### Grant Disclosures

The following grant information was disclosed by the authors:
Student Faculty Research Award to Rebecca Chastain and Benjamin Fitzpatrick by the University of Tennessee Graduate School.

### Competing Interests

The authors declare that they have no competing interests.

## Author Contributions

- Aidan Shaw conceived and designed the experiments, performed the experiments, analyzed the data, prepared figures and/or tables, authored or reviewed drafts of the article, and approved the final draft.
- Rebecca Chastain conceived and designed the experiments, performed the experiments, analyzed the data, authored or reviewed drafts of the article, and approved the final draft.
- Benjamin M. Fitzpatrick conceived and designed the experiments, performed the experiments, analyzed the data, prepared figures and/or tables, authored or reviewed drafts of the article, and approved the final draft.

## Animal Ethics

The following information was supplied relating to ethical approvals (*i.e.*, approving body and any reference numbers):

This work was approved by the University of Tennessee Institutional Animal Care and Use Committee protocol 2,710.

## Field Study Permissions

The following information was supplied relating to field study approvals (*i.e.*, approving body and any reference numbers):

Tissue collection in the Great Smoky Mountains National Park was approved by National Park Scientific Research Collection Permit GRSM-2023-SCI-2209.

## DNA Deposition

The following information was supplied regarding the deposition of DNA sequences:

The 270 cytochrome B sequences used in this work are available at GenBank (Table S1).

## Data Availability

The sequences and R code used to calculate Phi-st and perform the dbRDA are available in the Supplemental Files.

## Supplemental Information

Supplemental information for this article can be found online at http://dx.doi.org/10.7717/peerj.20110#supplemental-information.

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
