# Peer review of "Geographic distributions and patterns of co-occurrence among black-bellied and shovel-nosed salamanders (Desmognathus spp.) in the Great Smoky Mountains National Park"

_PeerJ, doi:10.7717/peerj.20110_

## Round 0.1 · original submission · Major Revisions

Thanks for your work to PeerJ.

Please change your manuscript as the comments from the reviewers.

·

Basic reporting

High quality for all aspects.

Experimental design

Experimental design is exceptional.

Validity of the findings

I have strong reservations regarding the validity of the cytochrome B sequences, particularly those for the Cades Cove lineage. Please see the annotated PDF.

Additional comments

Please see the annotated PDF for the referenced figures.

The authors provide an exceptionally sampled dataset, giving us a fine-scale resolution of the distribution of black-bellied and shovel-nosed Desmognathus in and around the Great Smoky Mountains National Park. It is phenomenal to see a resource like this, and I can only hope the authors (and other researchers) will continue such surveys in the future.

However, I have four major concerns regarding how the data are presented, how the phylogenetic dating was performed, how species boundaries are discussed, and whether the sequences for the new “Cades Cove” lineage are valid. The sequences included in the supplementary file do not appear to conform to the biological expectation of open reading frames and proper triplet counts for indels, nor do their patterns of nucleotide substitution match our expectation for mitochondrial dynamics in salamanders. I suspect that they might be nuclear-mitochondrial insertions from an ancient hybridization event between the Nantahala and Pisgah clades. It is imperative that the authors address these concerns of biological validity, although even if the sequences are nuMts, they would still likely be valid to show the patterns of geographic co-occurrence that are the primary focus of the MS.

One small note: it was very easy to overlook in their supplemental materials, but Pyron et al. 2025 did show that D. gvnigeusgwotli and D. mavrokoilus are in the same stream system near the Pigeon River outside of the NW margin of GRSM. RAP1686-9/AMNH A-195035-8 are D. gvnigeusgwotli from Stinking Camp Branch (35.731729 -83.058740), while RAP1690-1/AMNH A-195329-30 are D. mavrokoilius from Dogwood Flats Creek tributary (35.711306 -83.068138), all draining into the same stream leading into the Pigeon River.

1. Change the clade nomenclature
My first comment is to implore the authors to jettison or reduce the use of the Jackson 2005 A/B/C nomenclature, which will introduce substantial confusion into the literature and dramatically obscure and undermine the importance of the authors’ conclusions. I understand why they did this initially – to be internally logically consistent with the topological position of samples recovered from mitochondrial data as described previously by Jackson 2005 – but it creates monumental confusion when trying to cross-reference the valid species names, their phenotypes, their phylogenetic position, and their distributions.

For background, Jackson 2005 – in an unpublished MS thesis, introduced an informal A/B/C clade designation to refer to the unusual patterns of phylogenetic relatedness in black-bellied and shovel-nosed salamanders. This never became widely used.

Also in 2005, Kozak et al. 2005 introduced a widely used nomenclature using species names and A/B/C, such that D. amphileucus was – at the time – “quad/marm A,” while D. aureatus was “quad/marm B,” D. mavrokoilius and D. intermedius were “quad/marm C,” etc. This was expanded by Beamer and Lamb 2020 and has been used in dozens of publications, perhaps most prominently Pyron et al. 2020/2022, who demonstrated the species-level distinctiveness of these lineages.

Similarly, Jones and Weisrock 2018 introduced the valid and more euphonious and memorable “Nantahala” and “Pisgah” clade names for the divergent nuclear position of the northern and southern clades in the complex, which have also become widely used.

For the present authors to reintroduce the Jackson 2005 A/B/C nomenclature is a regression of more than 20 years of painstaking work to clarify the phylogenetic relationships and taxonomic status of these populations, as well as undermining the authors’ own stated goals from their abstract – “to clarify distributions and patterns of co-occurrence of recently described species in the Great Smoky Mountains.”

It seems imperative that the authors rely on the valid species names first and foremost in all descriptions of their results, to underscore the nature of the patterns they recovered. Use of “Nantahala” and “Pisgah” also seems desirable to clarify the distinct phylogenetic positions of these groups, which are obscured by the mitochondrial data due to known discrepancies revealed by phylogenomic analysis.

This can be seen most clearly in the figures, such as Figures 2 and 3. The illustration of ranges in Figure 2 is very misleading since it lumps multiple valid, described species together under clade names that can’t be connected to any of the recent phylogenetic, nomenclatural, or taxonomic work on these groups, while also failing to convey the relevant phenotypic information. For instance, “B & C together” dots really mean D. amphileucus, D. aureatus, and D. folkertsi, which are shovel-nosed and 2 black-bellied, while “Clade B” dots are two black-bellied species, D. amphileucus and D. gvnigeuswotli.

But those shovel-nosed Clade C dots are “quad/marm B,” while the Clade B dots are “quad/marm A” and “F,” and all are in the Nantahala Clade, while Clade A contains “quad/marm” C/D/E, otherwise known as D. mavrokoilius, D. kanawha, D. marmoratus, and D. intermedius in the Pisgah Clade, both black-bellied and shovel-nosed. Hopefully, it’s clear why this is an untenable nomenclatural system, as almost nobody could keep this straight. I’ve had an extensive discussion with Paul Super, the science coordinator at GRSM, having to explain at length why Jackson 2005 “marmoratus” in “Clade A” were actually “marm C” (i.e., D. intermedius) and not “quad A” (i.e., D. amphileucus) and that Clade C “marmoratus” were actually “marm B” (D. aureatus) in the Nantahala clade, not “marm C” D. intermedius! Please just use the species names.

This comes back in force for Figure 3, where a reader must attempt to hold in their minds what BB/SN and Clade A/B/C mean when interpreting the maps. Please spell out black-bellied and shovel-nosed, use the species names (D. gvnigeusgwotli and D. mavrokoilius for black-bellied and D. intermedius and “unknown lineage” for shovel-nosed), and provide a legend key that breaks them up by Nantahala and Pisgah. The colors in Figure 4 also don’t match those in Figures 2 and 3. It is burying the authors’ lede to not have figure 3 clearly drive home that “we found extensive co-occurrence of the black-bellied D. gvinigeusgwotli (Nantahala) and D. mavrokoilius (Pisgah) across the NW side of the Park – two distantly related but morphologically similar species which can apparently exist in sympatry.”

Even in the authors’ text it’s unclear, as they say things like “Both Jackson (2005) and Jones and Weisrock (2018) detected introgression between BB and SN ecomorphs in clade A but no evidence of introgression between ecomorphs in clades B and C (Nantahala), and it seems clear that there is no gene flow between northern (clade A) and southern (clades B and C) lineages (Pyron et al., 2020).“ This sentence shifts between calling it the Nantahala, “Southern,” or “B and C” clades, when Pyron et al. 2020 called it Nantahala, and not mentioning that Clade A is Pisgah. Please call them Nantahala and Pisgah for clarity.

2. Abandon or modify the dated analysis
The dating analysis presented is not sufficiently robust to support the level of confidence expressed by the authors and should be minimized or removed. First, 375bp of a single mitochondrial gene does not contain sufficient signal to parameterize a robust molecular-clock analysis. Second, the authors overlook the results of Budd and Mann (2024; Syst. Biol.) and Pennell (2023; Nature) regarding the insufficiency of these types of analyses for estimating species’ origins. Third and most importantly, the authors are violating the assumptions of the models by including population-level sampling across species. The prior densities and parametric models used in relaxed clock dating generally assume that tips are species, such that the distribution of substitutions along branches can be modeled as a Poisson process with a rate parameter describing interspecific variation.

Including multiple individuals within species violates this since the expectation of variable sites occurs with a much lower rate, inducing a mixture process in the posterior that isn’t accounted for in the prior. Consequently, the branch lengths, clock rates, and divergence times are distorted in artifactual ways that render them imprecise and inaccurate. This will be compounded immensely by the tiny amount of sequence data present. I realize that this strategy is very common, and the authors are only following many recent papers, but they are all generally suspect, and this strategy is generally untenable for dating divergences at the scale the authors propose here. Consequently, most of the “species ages” estimated by the authors are simply sitting midway from the root to the tip, as the posterior can’t meaningfully diverge from the prior. One thing the authors could do is sample the prior to demonstrate a shift, but if present, this would still not satisfy the issue of the prior assumptions being violated. I think the paper still stands fine without such an analysis.

3. Be more circumspect about species boundaries
I also take issue with how the authors present the discussion of species boundaries. They are laudably conservative in many instances, but I think they give the impression of too much interpretation of their limited mitochondrial data and an incomplete synthesis of the existing literature. As they note, all nuclear and mitochondrial data support essentially total isolation and firm species boundaries between the Nantahala D. amphileucus, D. aureatus, D. folkertsi, and D. gvnigeusgwotli, with no apparent gene flow between any of those species or with any lineages in the Pisgah clade.

Within the Pisgah clade, D. intermedius exchanges genes and mitochondria with geographically adjacent populations of D. mavrokoilius C (but not E or G), and D. marmoratus (not addressed in this study) likewise does so with D. mavrokoilius E and G, but not C. This is addressed in detail in Pyron et al. 2022, 2025. Finally, D. kanawha exhibits only limited past apparent gene flow with D. mavrokoilius, but not the shovel-nosed species. It is not accurate to say “Subsequently, Pyron et al. (2022) supported this interpretation with a larger dataset, but also presented evidence of mixed ancestry across all groups within clade A, consistent with extensive hybridization between ecomorphs.” We did not present evidence of mixed ancestry between all groups, but only a limited subset of geographically adjacent populations.

Similarly, the authors state, “Jones and Weisrock (2018) suggested that BB and SN ecomorphs in clade A could not be considered two distinct species, instead proposing genetic polymorphism or developmental differences as potential explanations for the two ecomorphs." However, they do not mention the extensive testing and compelling findings of Pyron et al. 2020 who showed that Jones and Weisrock 2018 did not actually sample any shovel-nosed individuals in their dataset and misidentified many specimens! The unpublished PhD thesis of Jones 2023 acknowledges this: https://uknowledge.uky.edu/biology_etds/92/
T
The authors then state that “This is consistent with random mating between clade A phenotypes within streams.” This is not true. Random mating within streams would produce small numbers of identical haplotypes shared between morphotypes, given their fast coalescence, but there is substantial variation in haplotype diversity that stretches across streams and between morphotypes. Furthermore, Pyron et al. 2022 conclusively demonstrated that most populations include individuals of “pure” ancestry (>80%) for each morphologically congruent lineage, which would be impossible if they were in panmixia in individual streams. Individuals with hybrid ancestry are a minority.

I realize the authors are trying to hedge when they say, “However, any single locus might be misleading, and our sample sizes are too modest to detect correlations between ancestry and phenotype much weaker than about 0.64 (Cohen, 1988). Thus, our data do not clearly differentiate whether clade A ecomorphs should or should not be classified as separate species, but do highlight the need for continued work in this system.” But I think this entire section is unnecessary and could be removed – the authors' data simply don’t permit speculation on species boundaries, especially given what is known about the extensive genealogical cohesion of these species from more extensive phylogenomic datasets.

The authors also state several times that their data reveal a high rate of hybridization, but this is not correct – these data cannot inform us as to the frequency with which hybridization takes place. What they reveal is that there is a high proportion of shared mitochondrial variation between the morphotypes. But these could have derived from a very small number of ancestral hybridization events and subsequent spread of captured mitochondria across populations with differential drift. This is already known from D. lycos and D. carolinensis (Beamer and Lamb 2020; Pyron et al. 2020, 2022).

Finally, while it is true that we said “These two BB species cannot be distinguished by any known visual characteristics (Pyron & Beamer, 2022),” this is not really accurate as presented – we meant diagnostically with certainty. It is quite easy to differentiate D. gvnigeusgwotli and D. mavrokoilius, even in the field. We provided Table 4, giving a range of qualitative color-pattern characteristics that can serve to differentiate them.

4. Further investigate the unusual sequences
The authors purport to discover a new Nantahala lineage of shovel-nosed in Cades Cove, going so far as to call it “sp. nov.” in Table 1. This is dramatically premature, as they offer no specimens, photos, measurements, vouchers, descriptions, etc. Using “sp. nov.” is usually paired with a description, but this is not given. More data would be needed before concluding that this is a distinct lineage, let alone a new species. In their tree, they don’t even show a “lineage” – implying monophyly – but a polytomy with multiple samples. Nuclear genomic data will be needed to clarify this, in comparison with existing datasets. They also don’t clearly explain why the Cades Cove shovel-nosed populations are a new lineage/species, but not the distinct clade of Smokies black-bellied populations that has approximately the same amount of divergence. However, this is not the main issue.

Unless I am making some mistake, the mitochondrial sequences presented for these specimens don’t appear to be valid cytochrome b sequences from a Desmognathus. I base this conclusion on several points.

First, the sequences as presented are not in an open reading frame from any codon position – any translation origin from 1 to 3 reveals internal stop codons.

Second, when aligned to other Desmognathus, there is a 10bp internal deletion induced, present in all other Desmognathus, which brings the sequences into “reading frame” before and after the indel, but it isn’t biologically plausible since it’s not divisible by 3. After aligning with outgroups (see below), this indel is absent from all batrachians. Finally, the 10bp frameshift/indel is found in all 255 of the GRSM sequences in the file provided by the authors, but they do not remark on this in the MS to provide an explanation.

Third, while the specimens come out as “sister” to D. aureatus + D. folkertsi in the phylogenetic analysis, they only appear to share 3–5 synapomorphies with those species, while exhibiting an astonishing ~30 autapomorphies – almost 10% unique, autapomorphic divergence from all other Desmognathus.

Fourth, they also exhibit 9.2% pairwise divergence with each other, in only 375 bp. This seems incredibly unlikely. This would require ~10% intrapopulation divergence between only 6 individuals sampled from 2 sites, much higher than known in most vertebrates or other desmogs – e.g., ~4–5% across all D. aeneus (Pyron et al. 2024; Mol. Ecol.).

Fifth, when BLASTed, their highest match is 84% similarity to D. folkertsi and 81–83% to D. aureatus, which would be exceptional amounts of pairwise divergence between sister species. The entire alignment of the other 22 Desmognathus species only has 13.7% pairwise divergence overall. The next-highest would be ~11% between D. valentinei and D. pascagoula. However, the Cades Cove samples also have 81% similarity to Hyla (OP344519.1), Mantella (AY263302.1), and Pleurodeles (DQ821206.1).

Making a quick ML tree with all of those sequences places the aureatus/folkertsi/Cades group outside of all Desmognathus, while removing them yields a monophyletic Nantahala clade nested within Desmognathus. The extreme branch lengths of the Cades samples are also clearly visible, greater than most interspecific branches in Desmognathus.

I can offer at least one explanation. What the authors hint at but don’t discuss directly is that the mitochondrial sister relationship between Nantahala and Pisgah – which is falsified by their distant positions in phylogenomic datasets – is thought to be the result of a deep-time reticulation and mitochondrial genome capture between the Nantahala and Pisgah lineages prior to the diversification of the extant species (Pyron et al. 2020, 2022, 2025). I think that the authors might have sequenced a combination of real mitochondria and pseudogenic nuclear-mitochondrial insertions from this reticulation. This would explain the exceptional divergence and lack of coding.

While this does not negate the value of their data for the intended purpose of mitochondrially genotyping sites for species occupancy, it requires a much greater explanation before we can trust both the topology and presence of a “new” lineage.

Reviewer 2 ·

Basic reporting

I find no issues with English, grammar, etc. throughout the paper, the writing style is certainly fine. However, there are a total of 41 references cited in the paper (2 of which are not referenced in the paper, see my comment below), 7 of which are computing or map programs or general lab protocols. Neither the intro. Nor does the discussion (although the discussion comes far closer) place this study within the context of the current state of the field and our understanding of salamander evolution in general and the fascinating field of desmog evolution in particular. The authors do cite previous studies, but in my opinion, they provide a cursory treatment and miss an opportunity to provide a thorough review of the state of this field. If this work provides a significant contribution, this is necessary. 

I have pasted a specific comment here regarding the introduction illustrating my comments above:

Lines 51 – 63 – In this entire paragraph, only two sources are cited. This is a scientific paper, so unless you are the first to discover all these observations, you must provide the reader a review of at least a good portion of the literature that came before. I do not think a paragraph such as this should appear in any scientific paper. However, simply from the author’s perspective, you want to provide a quality product that both protects your reputation as a scientist and produces a work that provides benefit to the field and will be utilized by other scientists.

This is always difficult, but there are cases where references are cited in the manuscript that do not appear in the literature cited (Pyron et al. 2025) and where references appear in the literature cited that do not appear in the manuscript (Sambrook and Russell 2001, QGIS). There certainly might be more, I simply noticed these three during my review.

The maps, at least Figure 3, need work. I have pasted several specific comments regarding them below:

Line 174 – “…these six SN, three were sampled from Mill Creek at the western end of Cades Cove and three from Anthony Creek at the eastern end (see Fig. 3).”

Cades Coves does not appear in Figure 3 and needs to be added if it is referenced in this manner.

Also, regarding figures, the photos in Figure 1 are amazing, the maps a good but less impressive/borderline professional appearing. QGIS is cited, but never mentioned in the manuscript (an issue), so apparently it was used to make these. I struggle greatly with QGIS, but it can produce a really great product. I suggest consulting someone for assistance to improve these a bit.

Line 307 and following paragraph – This entire paragraph refers to stream names and park locations that the reader, unless they happen to frequent the park, will have no knowledge of. These need to be included in the map for this paper to appeal to more than a local audience.

Experimental design

The research question is well defined and of great interest to many, both those studying these specific salamanders and ecologists in general. However, I do have some concerns regarding the methods.

My primary concern is that the reader (or reviewer in this case) has been provided insufficient information to judge the quality of the data, and several aspects of the manuscript are a bit disconcerting. The premise of the study is the construction of phylogenies based on a 375 bp mit. region. Clearly, the conclusions rest on the quality of these sequences. While the acquisition of these in a manner that facilitates complete confidence should have been a simple task, the reader is left a bit uncertain about this, as the manuscript now reads.

The following are the complete methods for the post-extraction molecular techniques:

We performed PCR to amplify approximately 500 bp of cytochrome B using the primers and protocol described by Jackson (2005). PCR products were purified using Exo-I-SAP and sequenced by Eurofins Genomics.

The entire paper rests on the quality of these sequences, but the reader has no idea how these 289 reactions were performed. I have pasted my specific comments regarding these methods below.

Line 126 and following – In my opinion, this description of the acquisition of the cytb sequences, the entire premise of the paper, is acceptable for an email to a colleague describing what was done (a total of six lines, four sentences), but not acceptable for any scientific journal. I would be incredibly frustrated to come across this paper with molecular methods in this state. Details are given below.

Line 127 – Why two extraction methods? Ideally, we would like to know how many were extracted each way, but certainly why you used two. And what were the yields like for each (Qubit or Nanodrop)? How long did you incubate the tissue (at least for the DNeasy kit)? Overnight, a few hours? As a scientist doing similar work (your target audience here), I would like to see what worked for you as I scan these methods before reaching the results. This must be added before publication and greatly enhances the value of this work.

Line 130 – It is necessary to include your primer sequences, reaction volumes, reaction parameters, thermocycler type, and most importantly, the components utilized, including the Taq used. It’s likely that some of these parameters varied slightly, particularly because the methods cited are found in a 20-year-old dissertation and are highly antiquated (they used a Perkin-Elmer 9600 thermocycler). Although it appears this dissertation is readily available, it is time-consuming for the reader to find these items there. It greatly increases the utility of your manuscript to include these items briefly within the manuscript and fully within the supplemental material. It’s an easy fix.

Regarding the Taq, this does not appear to be clear in Jackson 2005. The Taq is simply described as “Taq.” In this case, the Taq utilized is incredibly important, presumably, it was high fidelity Taq of some sort. If not, the reactions likely should have been run in duplicate (not both ways, but using two separate PCR reactions). If the hi-fi taq was not used and the reactions were not done in duplicate, I do believe the work can proceed, but this is a major issue that is going to require extensive explanation and justification based on the literature. In any case, as currently written, the reader has no idea what was done here, so no reference for how much confidence they should have in these sequences.

Line 132 – How were sequences viewed and trimmed? MegaX, Geneious, BioEdit?

Line 133 – “We then combined our sequence data with sequences generated by Jackson (2005) and performed phylogenetic analyses using two Bayesian inference models.” The reader has no idea how many samples were provided by this study (although that is mentioned in the results) and how many were provided by Jackson (2005) for the analysis. This must be clearly stated as it is the premise of the entire study.

Line 301 – “Some samples with low-quality reads could be assigned confidently to major clades using BLAST but were not included in our gene tree estimates.”

For me, this is one of the biggest concerns of this work: “samples with low-quality reads.” The premise of the work is building phylogenies based on slight bp differences in a short 375 bp mt region. There is absolutely no reason any of the sequences used in this study should be of low quality. This is the simplest of PCR reactions (for which we were provided no details regarding methods), and, if a HiFi Taq was used (see my earlier comments), the authors should have near complete confidence in all their sequences.

Line 303 – “Of the 255 individuals included in our phylogenetic analysis…”

This is especially disconcerting. The reader is left to assume that of 289 samples, 34 (289-255) or 12% were “low quality.” Why even one would have been of low quality (anything low quality should have simply been redone) remains a question. But the reader is supplied with no information regarding the criteria for “low quality,” or more importantly, the criteria for sufficient quality for the remaining samples.

As I noted previously, this reference appears in the lit. cited but is never referenced in the manuscript. I am uncertain what the authors had in mind; if they were citing it for the simple PCR they ran, this is an interesting choice. - Sambrook J, Russell DW. 2001. Molecular cloning: a laboratory manual. Cold Spring Harbor, 434 N.Y: Cold Spring Harbor Laboratory Press.


Additionally, the authors use the term "trout stream," which is odd and raises a number of questions. I have pasted specific comments below:

Line 196 – The use of the word “trout streams” absolutely must be changed. Rainbow trout, brook trout, or both occur in every lotic system in the park and, along with black nose dace, are the last fish to disappear as first-order streams become smaller and smaller with increasing elevation. They persist at increasing elevation in the smallest of first-order streams. Use of the term “trout stream” is meaningless in the park; every lotic system is a trout stream.

This needs serious work and the addition of objective variables to the analysis. If none were taken in the field, then two that can be added retrospectively are elevation and stream order. This must be completed if this manuscript is to move forward.

Validity of the findings

Assuming the quality of the sequences used to generate these conclusions was sufficient (see previous section), I believe this study makes a significant and novel contribution to the field. Although the discussion found in the conclusions is better than that in the introduction, as I mentioned previously, I do believe it needs to be expanded to allow the reader to appreciate these findings within the context of both salamander evolution in general and demog evolution in particular.

Additional comments

Introduction
Line 46 – “contains the entire Great Smoky Mountains range…” – This appears to be incorrect, either way a citation needs to be provided to support this statement.

Line 47 and following – “but the number and identity of the species occurring there is uncertain after recent taxonomic revisions of the genus Desmognathus (Pyron & Beamer, 2022, 2023). – This needs to focus specifically on the taxonomic changes in this subset of Desmog species. The number of salamander species in the park prior to this revision needs to be mentioned (32?), and the number of Desmog species prior to these revisions (8?) discussed. The manuscript will be greatly improved by rewriting this paragraph, and the rest of the introduction to this point, to provide a review of the existing literature that provides a rational for the study.

Line 51 – “(BB and SN, hereafter)” - If PeerJ likes the use of “hereafter” I don’t object. I no longer see this used in modern manuscripts and to me it seems redundant and unnecessary.

Line 57 – “Shovel-nosed salamanders are of comparable size to BBs.” – But two lines earlier it was stated that BB’s are the largest, just need to clean this up a little. Are SN slightly smaller, either way a reference needs to be provided here.

Line 94 – “Here we use expanded sampling from across the Great Smoky Mountains.” – “From” needs to be removed. Also, why the sudden drop of “GSMNP?” The first line of the methods and Figure 3 indicate all samples were collected inside the park. If that was the case this needs to be changed.

Line 93-94 – “including some large population samples” – What does this mean? You collected a larger number of individuals at site? This needs to be reworded.

Methods

Line 104 – The manuscript would be improved by providing specific numbers on how many specimens were collected via the three methods listed: 1) visiting targeted sites, 2) opportunistically at seeps, 3) electrofishing with park personnel. I consider this mandatory. I also recommend adding minimal detail regarding the electrofishing – clearly it was backpack, DC, IBI or three-pass?

Line 114 – “Due to our non-invasive sampling technique and the efficiency of electrofishing, we usually sampled every individual caught, except for very small larvae (less than 2.5 cm total length) and individuals with prior tail damage.”

This is an interesting sentence for two reasons – I’ve never heard of electrofishing referred to or thought of it as non-invasive. I suppose it is relative to rotenone, but after spending a good bit of my professional life conducting it, I think it would be challenging to consider it as such (involves wading through the creeks, some fish do perish, all captured fish are temporarily removed from the creek, etc.).

Second, what do either of these (non-invasive tech. of EF efficiency) have to do with sampling every individual caught (note the “due to” at the start of sentence)? So if it was an invasive technique, or EF was less efficient, you would not have been able to sample every individual? It seems like in the second case it would be have been more likely you would have been able to do so. In any case, this appears to require restructuring.

Line 119 – I don’t recall ever seeing an entire IACUC in the midst of a manuscript. If the journal prefers that then certainly it should remain. It does appear odd.

Line 139 – I recommend explaining why Desmognathus wright was selected for use here.

Line 145 – This appears to be mixed in the literature. I would have thought “c” in clade C (and all specific references to clades) should be capitalized but perhaps not.
Results

Line 155 – This is the first mention of number of samples collected, this must be placed in methods. We really need some indication of the number of individuals collected at each site, that would be easy to add to Figure 3.

Line 164 – Again, I am very uncomfortable with the combining of these data to produce the results that are the entire premise of the paper. I would like to know how many samples of each species came from each study and an overview of Jackson’s methods. Was that thesis/dissertation never published? We assume not. In any case, since the results of the study rely on an unknown percentage of Jackson’s data I think we need to know more about the methods from that study, beyond what is provided in the introduction.

Line 180 – “One sample from Cades Cove was uncertain in its relationship to other Cades Cove samples.” I would add which side of the cove this sample occurred, since you’ve already mentioned that.

Additionally, referencing Table 1 here appears to be a mistake. There is no mention of locality of any type in Table 1.

Line 198 – “This highlights the pattern that BB are less dependent on trout streams than SN and more likely to be found in smaller seeps and trailside.”

This is discussion but appears in the results. This assertion must be supported by data in a table or figure, not just a statement by the authors that this was what was observed. Am I missing the presentation of these data? I am unable to find any indication of habitat types in which each clade was collected. Has the wrong table been inserted where Table 1 should go?

Line 202 and following – Again, if these areas in park are referred to by referencing Figure 3 they need to appear in Figure 3.

Discussion

Line 227 – Need a reference, or better yet multiple, for this. Easy to obtain, cite Dodd and others.

Line 238 – “These findings emphasize the value of parks for preserving biodiversity, both known and not yet described.”

This is an oddly placed statement. While true, it is only relevant to this study in the broadest sense and has not been mentioned since the introduction. The work would be strengthened by focusing here on tying together the preceding paragraph and saving this for the end of the discussion, if it’s included at all.

Line 253 – Again “trout streams.”

Line 270 – No Pyron et al. 2025 appears in the literature cited.

Line 298 – This is not the correct use of the phrase “DNA assay.” Perhaps “our analysis of a 375 bp mitochondrial region” or something similar.

---

## Round 0.2 · Minor Revisions

·

Basic reporting

The authors have responded in detail to all major comments. The manuscript is well-written, properly cited, properly structured, and self-contained.

Experimental design

As in the previous review, all experimental design criteria are satisfied.

Validity of the findings

Data are provided, apparently sound, and replicable. Conclusions are logical and valid.

Additional comments

The authors have done an excellent job revising the MS. I am satisfied by the validity of the sequences, the novelty of the Cades SN lineage, and share their doubt about nuMts, which don't seem to be present in the fuscus genome. All other aspects were handled well.

My only remaining comments are:

1) I previously forgot to mention that Pyron and Beamer (2022) also reported a Pisgah/Nantahala sympatry site in Hickory Nut Gorge (p. 18, Fig. 9) between amphileucus and mavrokoilius.

2) On line 82, the authors say, "This pattern supports the hypothesis that co-occurrence and divergence of BB and SN ecomorphs has been facilitated by ecological niche partitioning."

The authors curiously make no mention of the previous papers by Voss et al. 1995 and Jones et al. 2006 who point out the strong influence of ancient and current river drainages in structuring genetic diversity in the highly aquatic SN species. Indeed, this is the primary determinant of geographic species boundaries for the three described species, making the Cades Cove SN particularly astonishing.

I would suggest that the authors cite and briefly discuss this aspect in the final version of the paper:

Jones, M.T., Voss, S.R., Ptacek, M.B., Weisrock, D.W. & Tonkyn, D.W. (2006) River drainages and phylogeography: an evolutionary significant lineage of shovel-nosed salamander (Desmognathus marmoratus) in the southern Appalachians. Molecular Phylogenetics and Evolution, 38, 280–287.

Voss, S.R., Smith, D.G., Beachy, C.K. & Heckel, D.G. (1995) Allozyme variation in neighboring isolated populations of the plethodontid salamander Leurognathus marmoratus. Journal of Herpetology, 29, 493–497. https://doi.org/10.2307/1565011

---

## Round 0.3 · accepted · Accept

Considering the evaluations of the three reviewers, I find the manuscript suitable for publication. The corrections and clarifications provided have improved the manuscript and adequately addressed the reviewers’ initial concerns. The authors should, however, make minor adjustments: address Reviewer 1’s two additional comments, add Dodd (2004) to the reference list, and check the accessibility of the basemap link during the proof stage. Apart from these small revisions, I see no need for further external review and am pleased to recommend the manuscript for publication.

·

Basic reporting

-

Experimental design

-

Validity of the findings

-

Additional comments

I have only two tiny comments:

Line 265 "Based on the same data..." It was not the same data. Pyron et al. (2022) used anchored hybrid enrichment (target capture), and Pyron et al. 2025 used genotype-by-sequencing (RADSeq).

Several times, the authors cite Jones et al. (2005) when they mean Jones et al. (2006). The reference is listed correctly as 2006.

Reviewer 2 ·

Basic reporting

Following completion of a review of the author’s revisions and the new supplemental methods, I find the authors have done a good job addressing concerns with the original manuscript. References have been added, many portions of the manuscript have been rewritten, sequences of questionable quality have been omitted, and clarifications have been implemented. The quality of the maps has also been somewhat improved. As I stated in my original review, I believe this study makes a significant and novel contribution to the field. I now find the manuscript suitable for publication in its current form.

Experimental design

No new input.

Validity of the findings

Please see my previous statement.

Reviewer 3 ·

Basic reporting

-

Experimental design

-

Validity of the findings

-

Additional comments

Thank you for the invitation to review the manuscript entitled “Geographic distributions and patterns of co-occurrence among black-bellied and shovel-nosed salamanders (Desmognathus spp.) in the Great Smoky Mountains National Park” for PeerJ. I would like to express my appreciation for the relevance of the study on the Desmognathus quadramaculatus–marmoratus complex in the Great Smoky Mountains. The manuscript stands out for its valuable contribution to our understanding of cryptic diversity by integrating molecular tools with state-of-the-art Bayesian phylogenetic analyses.
The identification of new zones of sympatry among morphologically cryptic species and the discovery of divergent mitochondrial lineages significantly enhance our knowledge of the biogeography and community structure of these amphibians. The detection of a lineage related to D. aureatus in an area where it had not previously been recorded underscores the importance of continued systematic exploration using integrative approaches.

I believe this study represents a significant contribution to both the systematics and conservation of herpetofauna. Its publication in PeerJ will help to highlight the importance of preserving these unique ecosystems and promote further high-quality research in the area.